# Study on Damage Characteristics of Water-Bearing Coal Samples under Cyclic Loading–Unloading

**Hongxin Xie, Qiangling Yao \*, Liqiang Yu and Changhao Shan**

School of Mines, China University of Mining and Technology, No. 1 University Rd., Xuzhou 221116, China;
xhx1998xhx@163.com (H.X.); yuliqiangcumt@163.com (L.Y.); shanchanghao_cumt@163.com (C.S.)
* Correspondence: yaoqiangling@cumt.edu.cn

**Abstract:** For underground water reservoirs in coal mines, the complex water-rich environment and changing overburden stress can damage coal pillar dams. In this paper, the coal samples from coal seam $2^2$ of Shangwan coal mine were taken as research objects and the damage mechanism and characteristics of coal samples with different moisture content and wetting-drying cycles under cyclic loading were investigated. The results show that as the moisture content and wetting-drying cycles increase, the post-peak stage of the coal samples under cyclic stress becomes obvious, and the hysteresis loop changes from dense to sparse. Compared to the uniaxial compression experiment, when $w$ = 5.28% (the critical water content), mechanical parameters such as peak strength and modulus of elasticity decrease the most. Under cyclic loading, the damage mode of both sets of coal samples was tensile damage, but the increase in wetting-drying cycles promotes the development of shear fractures. For evaluating fracture types, the RA-AF density map is more applicable to wetting-drying cycle coal samples, whereas for the coal samples with different moisture contents this should be carried out with caution. This study can provide some theoretical basis for the stability evaluation of coal pillar dams in underground water reservoirs.

**Keywords:** moisture content; dry–wet cycles; cyclic loading; injury mechanism; acoustic emission RA-AF



## 1. Introduction

Western China is rich in coal resources but short of water resources. The uncoordinated distribution of coal and water has become an important factor restricting the realization of green mining in Western China [1,2]. To solve the contradiction of coal and water co-mining, the construction of underground reservoirs in coal mines has been put forward to solve the water shortage in arid and semi-arid mining areas in Western China [3,4]. However, the coal pillar dam of the reservoir will be damaged or made unstable under the joint action of water immersion and periodic pressure of overburden [5]. Therefore, it is important to study the deformation, failure characteristics, and instability precursor information of coal samples with different moisture content under cyclic compression loads.

In recent years, research on water–rock interactions has been widely performed. By studying the micro-structure and adsorption behavior of clay minerals, Cherblanc et al. concluded that the weakening of the mechanical strength of water-bearing limestone is related to the adsorption capacity and that the clay delays the weakening of the rock [6]. Moreover, Chen et al. found that the weakening of mechanical factors such as the compressive strength of rock materials after water damage was related to changes in the micro-structure by SEM [7]. The results have shown that with the increase in moisture content, rock samples present more significant plastic characteristics, whereas its uniaxial compressive strength and elastic modulus decreased to varying degrees [8,9]. Through the analysis of the mineral composition of "Macigno" sandstone, the water absorption and water saturation are increased with the increase in montmorillonite content, which may play an important role

in the decay process of sandstone [10]. Bensallam et al. studied the mechanical behavior of clay under the alternation of dry and wet, and found that the deformation of expansion and shrinkage decreased by the alternation of dry and wet, and the change of soil behavior was affected by load-deformation [11]. Yao et al. first studied the weakening mechanism of moisture content and different soaking times on the mechanical properties of coal rock [12]. Huang et al. studied the effects of dry–wet cycles on the mechanical properties of sandstone and mudstone and found both the elastic modulus and uniaxial compressive strength of sandstone and mudstone were reduced by dry–wet cycles, and the degradation rate of the two mechanical parameters of mudstone was always larger than sandstone [13]. At the same time, water and temperature have multiple weakening effects on the mechanical properties of coal measures mudstone. Through creep experiments, it is found that high temperatures will aggravate the weakening effect of water [14].

Moreover, research on the mechanical properties of coal rock under cyclic loading has been also carried out [15,16]. For porous sandstone, the two damage mechanisms of compaction and micro-cracking under cyclic loading are affected by the dip angle of isotropic plane load, which decreases gradually with increasing confining pressure [17]. In addition, the residual strain method and the axial secant modulus method could describe the initial fatigue damage and degradation process of sandstone samples [18]. Through scanning an electron microscope, Erarslan et al. found that the failure of the Brisbane tuff is inter-granular fracturing and trans-granular fracturing, which may be due to frictional sliding within the weak matrix [19]. Combined with the experimental study of deformation and damage of fine sandstone under cyclic loading, Jia et al. found the change rate of a transverse and longitudinal strain first increases and then remains unchanged. With the increase in cyclic stress levels, the failure mode changes from compaction to expansion [20]. Zhang et al. applied the Single-link cluster (SLC) method to the spatial and temporal evolution of acoustic emission and the damage evolution process of coal samples, confirming the effectiveness of the SLC method [21]. A large number of studies on the damage to coal under cyclic loading show that the elastic modulus increased at first and then decreased with the number of cycles, the peak stress increased step by step, and the damage to coal was related to the level of cyclic stress [22,23].

The above studies mainly focus on the damage characteristics of coal rock with different water-bearing conditions under the uniaxial compression, as well as the mechanical properties and AE characteristics of coal rock under cyclic loading–unloading tests. However, research on the mechanical properties and fracture damage of water-bearing coal samples under cyclic compression load has been rarely carried out.

In this study, the self-designed, non-destructive water immersion equipment [24] was used to conduct non-destructive water immersion and dry–wet cycle treatment on coal samples [25]. Coal samples with a moisture content of 0%, 3.95%, 7.46%, and 15.25% were marked as Experimental group A, and those treated by dry–wet cycles of 1, 2, and 3 were marked as Experimental group B. The MTS universal servo testing machine was used for the mechanical experiment, and real-time photography and an acoustic emission system were used for monitoring during the experiment. In this study, the mechanical properties and damage mechanism of coal samples with different moisture contents under the cyclic compression load were mainly studied. The research results are important for understanding the failure mechanism and precursory information of coal pillar dams under the action of moisture content and cyclic compression load.

## 2. Materials and Methods

### 2.1. Test Materials and Specimen Preparation

Shang-wan Coal Mine is in EjinHoro Banner, Ordos City, Inner Mongolia Autonomous Region. It belongs to the western mining area and is short of water resources. If the mine water is not stored, a large amount of water resources will be drained and lost on the surface, which will aggravate the ecological damage on the surface and cause a water shortage in the mine and surrounding areas. Therefore, Shang-wan Coal Mine has implemented

the underground reservoir technology in the $2^2$ coal gob, that is, through the connection between the artificial dam and the protective coal pillar, the mine water is sealed in the gob to form a storage dam to protect the mine water resources.

As shown in Figure 1, the underground reservoir diagram shows that the coal pillar dam is not only in the complex stress field affected by mining stress, the lateral stress of caving gangue, and stagnant water pressure in the gob, but also affected by the long-term erosion of stagnant water in the gob, resulting in dynamic changes in the water content of the coal pillar. In addition, the repeated scouring effect of water makes the coal pillar in a dry-saturated water-bearing state. Considering the stress redistribution under the influence of mining and the long-term erosion of water to the coal pillar dam, an experiment is designed to study the long-term stability of the coal pillar dam.

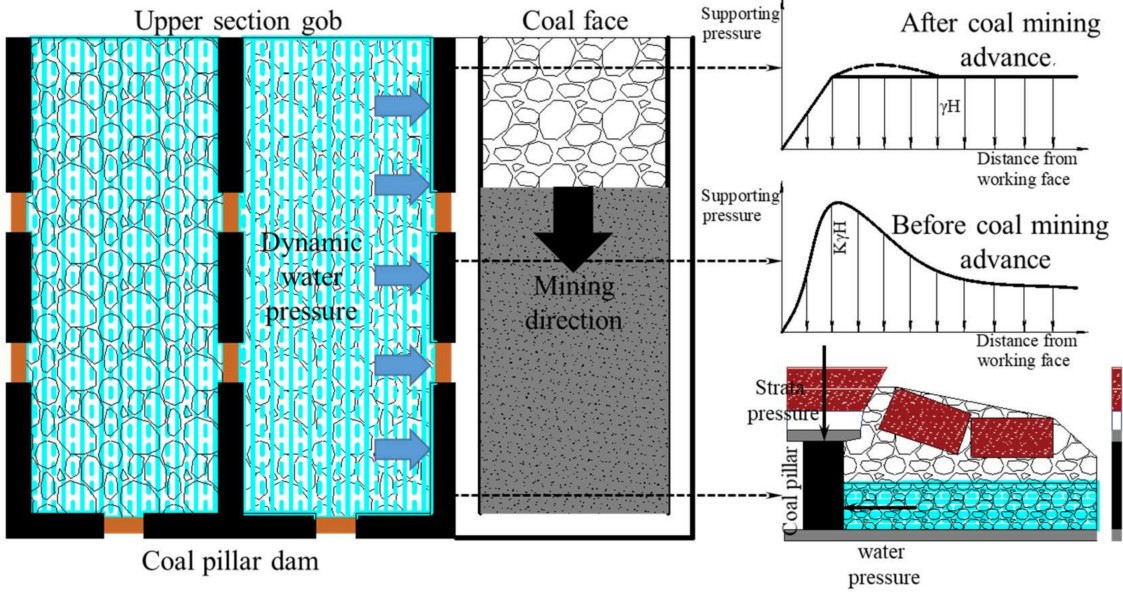

**Figure 1.** Schematic diagram of underground reservoir.

In this test, coal samples were taken from the coal seam $2^2$, as shown in Figure 2. The moisture content of coal samples in the natural state was 7.59%. According to the requirements of the Test Code of the *International Society of Rock Mechanics* [26], the raw coal was processed into 30 standard cylindrical samples of 50 mm × 100 mm. The presence of bedding in the coal samples has an effect on deformation damage and permeability evolution [27,28]. In order to eliminate the influence of the bedding layer on this cyclic loading and unloading experiment, we carried out a wave velocity test before preparing the specimen to ensure the homogeneity of the specimen as much as possible [24].

The samples were divided into 7 groups: there are 4 groups of samples with different moisture contents (Experimental group A: 0%, 3.95%, 7.46%, and 15.25%) and 3 groups of samples treated by different dry–wet cycles (Experimental group B: 1, 2 and 3 cycles). These samples were numbered in the form of Experimental group No.—Moisture content/dry–wet cycles—Number. For example, A-1-1 represents the first coal sample with the moisture content of 0% in Experimental group A, and B-1-2 represents the second coal sample with one dry–wet cycle in Experimental group B. The uniaxial compression experiments and cyclic loading–unloading uniaxial compression experiments were carried out. Table 1 shows detailed information about the samples. Refer to Equation (1) for calculating moisture content.

$$w = \frac{M' - M_{dry}}{M_{dry}} \times 100\% \tag{1}$$

where, $M'$ is the mass of the characteristic water-bearing coal samples, and $M_{dry}$ is the mass of the drying samples.

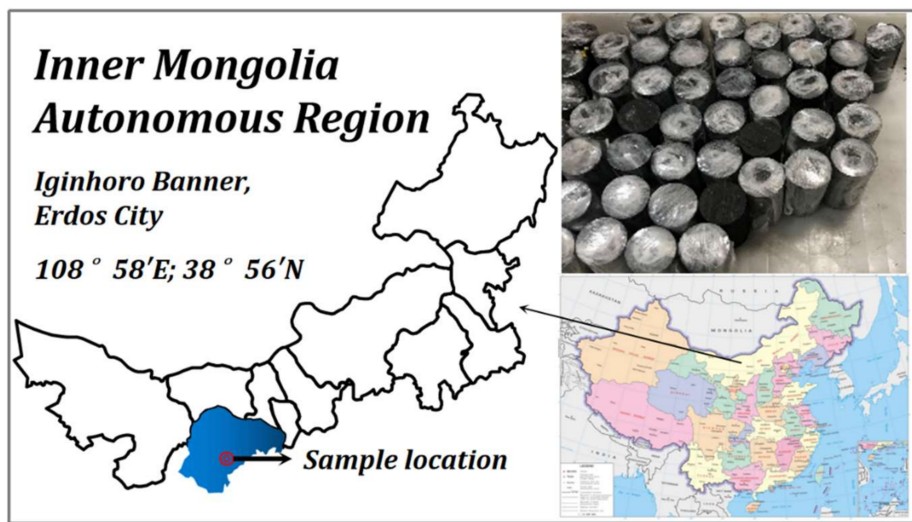

**Figure 2.** Sampling location.

**Table 1.** Sample number.

| Sample No. | A-1 | A-2 | A-3 | A-4 |
|---|---|---|---|---|
| Moisture content | 0.00% | 3.95% | 7.46% | 15.25% |
| Table | A-1-1~A-1-3 | A-2-1~A-2-3 | A-3-1~A-3-3 | A-4-1~A-4-3 |
| Table | B-1 | B-2 | B-3 | |
| Immersion times | 1 | 2 | 3 | |
| Table | B-1-1~B-1-3 | B-2-1~B-2-3 | B-3-1~B-3-3 | |

*2.2. Experimental Scheme and Equipment*

The experiment was divided into two parts: (1) uniaxial compression experiment and AE experiment of coal samples with different moisture contents. The variation laws of mechanical parameters (such as stress, strain, and AE characteristics) of coal samples with different moisture contents and different dry–wet cycles at a constant loading rate were measured. Based on this, the initial loading strength value of cyclic loading–unloading was set. (2) In the second part, the uniaxial cyclic loading–unloading experiment and AE experiments of coal samples with different moisture contents were carried out, and the variation laws of mechanical properties and AE characteristics of coal samples with different moisture contents under cyclic loading–unloading were obtained and compared with those under the ordinary uniaxial loading mode.

As shown in Figure 3a, the wave velocity of the specimen was tested using an ultrasonic velocimeter (Model RSM-SY6, Wuhan Zhong Yan Technology Co., Ltd., Wuhan, China). As can be seen in Figure 3b, the wave velocity of the coal sample was distributed in the interval of 1425–1520 $m \cdot s^{-1}$, with an average wave velocity of 1477 $m \cdot s^{-1}$, and the variance and skewness were 0.6 and −0.2, respectively, with a small dispersion. This indicates that the internal structure and defect distribution of the specimens are consistent, which can reduce experimental errors.

The 101-2 electric constant temperature drying oven was selected as the drying equipment, as shown in Figure 4a. According to the requirements of the Test Code of the *International Society of Rock Mechanics* [26], the drying temperature was set at 110° and the drying duration was set at 12 h. After drying, the coal sample was taken out and wrapped tightly with plastic wrap in time to prevent its water absorption in the natural environment from affecting the accuracy of the experiment. The self-designed HL-8-1WS rock moisture content continuous weighing detector [29] was used to immerse coal samples, as shown in Figure 4b. The coal samples were in full contact with the saturated humidity water vapor in the constant temperature and humidity box to absorb water freely. In this way, the damage

to the coal sample caused by excessive direct immersion water pressure was avoided, and the integrity of the coal sample was protected to a great extent.

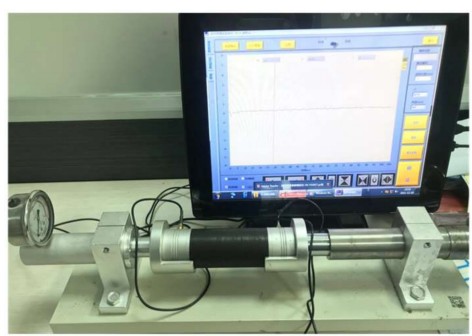

(**a**) Ultrasonic Velocimetry

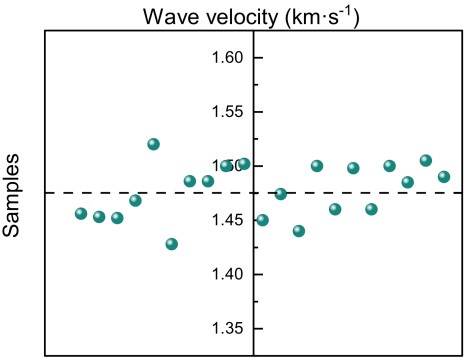

(**b**) Specimen wave velocity

**Figure 3.** Specimen wave velocity testing.

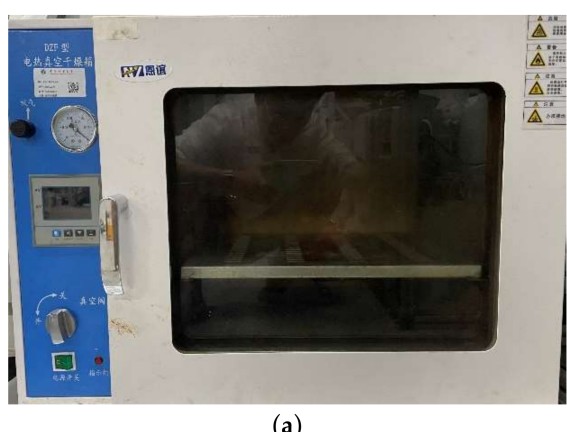

(**a**)

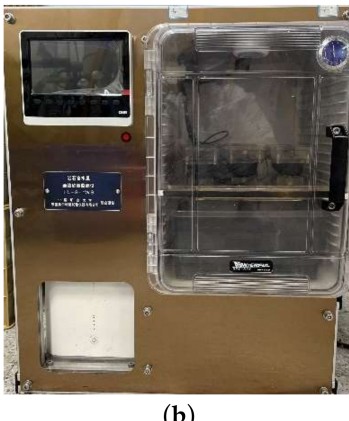

(**b**)

**Figure 4.** (**a**) Electro-thermal constant temperature drying oven; (**b**) Continuous weighing detector for rock water content.

The MTS electro-hydraulic servo universal testing machine in the State Key Laboratory of Coal Resources and Safe Mining, China University of Mining and Technology was selected as the loading system. The displacement loading at a rate of 0.3 mm/min was employed in the uniaxial compression and cyclic loading–unloading uniaxial compression experiments. The difference was that in the latter experiment, 50–70% of the expected peak strength (which was obtained from the uniaxial compression test) was first loaded and unloaded to 1% of the expected peak strength and added with a gradient loading (taking 10% of the expected peak strength as a gradient), and then repeated until the coal sample was damaged. At the same time, the PCI-II acoustic emission system produced by the American Acoustic Physics Company (PAC) was used to monitor the acoustic emission signals during the experiment. Referring to the previous study conducted by Xia et al. [30], a total of 4 probes were arranged at the upper and lower 1/3 points of the 4 surfaces of the coal sample cylinder (2 probes on the upper and lower horizontal planes, respectively), with a distribution of 90°, to ensure the accuracy of acoustic emission monitoring. During the test, the Nikon Z6II high-definition SLR camera was used to monitor the loading process and record the fracture failure mode of coal samples during compression. Figure 5 shows the test system and the load loading path.

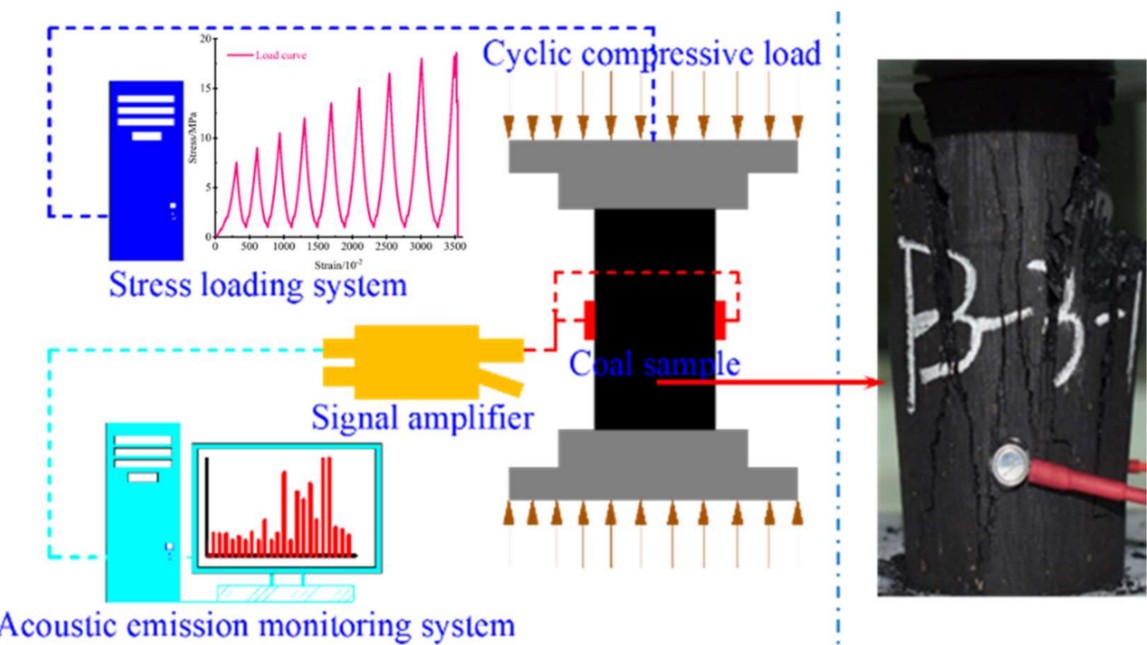

**Figure 5.** Schematic diagram of rock mechanics and acoustic emission test system.

*2.3. Acoustic Emission RA-AF Relational Density Method*

Acoustic emission *RA* value (Rise time/Amplitude) can be used to characterize the failure mechanism of coal rock, as well as the precursor information to predict the fracture development and failure of coal rock [31,32].

Reference *RA* value definition [33]:

$$RA = \frac{RiseTime(RT)}{Ampltitude(Amp)} \tag{2}$$

where *RT* is the rise time of the AE waveform, in us, *Amp* is the amplitude, in dB. If the amplitude is given on a logarithmic scale (for example, dB), it should be converted to the original voltage. The AE pair *V* is solved as follows, with the result expressed in volts.

$$A(d\text{B}) = 20 \times \lg\left(\frac{V}{V_{ref}}\right) - G \tag{3}$$

where *G* is the AE amplification gain (threshold value), in this experiment *G* = 40 dB, $V_{ref}$ is the reference value used in AE software is 1 μv.

The Equation (4) for calculating the *RA* value expressed by the original voltage *V* can be obtained by the deformation of Equation (3).

$$V = V_{ref} \times 10^{\frac{V}{V_{ref}}} - G \tag{4}$$

The *AF* value is defined as:

$$AF = \frac{Ringcount(\text{AE})}{Duration(\mu s)} \tag{5}$$

In order to reflect the RA-AF distribution characteristics more intuitively, the RA-AF probability density distribution maps of coal samples with different moisture content were made by using MATLAB R2022a. The color evolution of the AE data density allows a clearer qualitative analysis of the loss evolution of the experimental specimen. The red area

represents the maximum data density, the blue area represents the minimum data density, and the white dashed line is the line dividing the pulling shear cleavage.

## 3. Results and Discussion

### 3.1. Mechanical Properties of Water-Bearing Coal Samples under Uniaxial and Cyclic Loading–Unloading Experiments

Figure 6 shows the stress curve and mechanical parameter characteristics under the uniaxial compression tests and cyclic loading–unloading uniaxial compression tests.

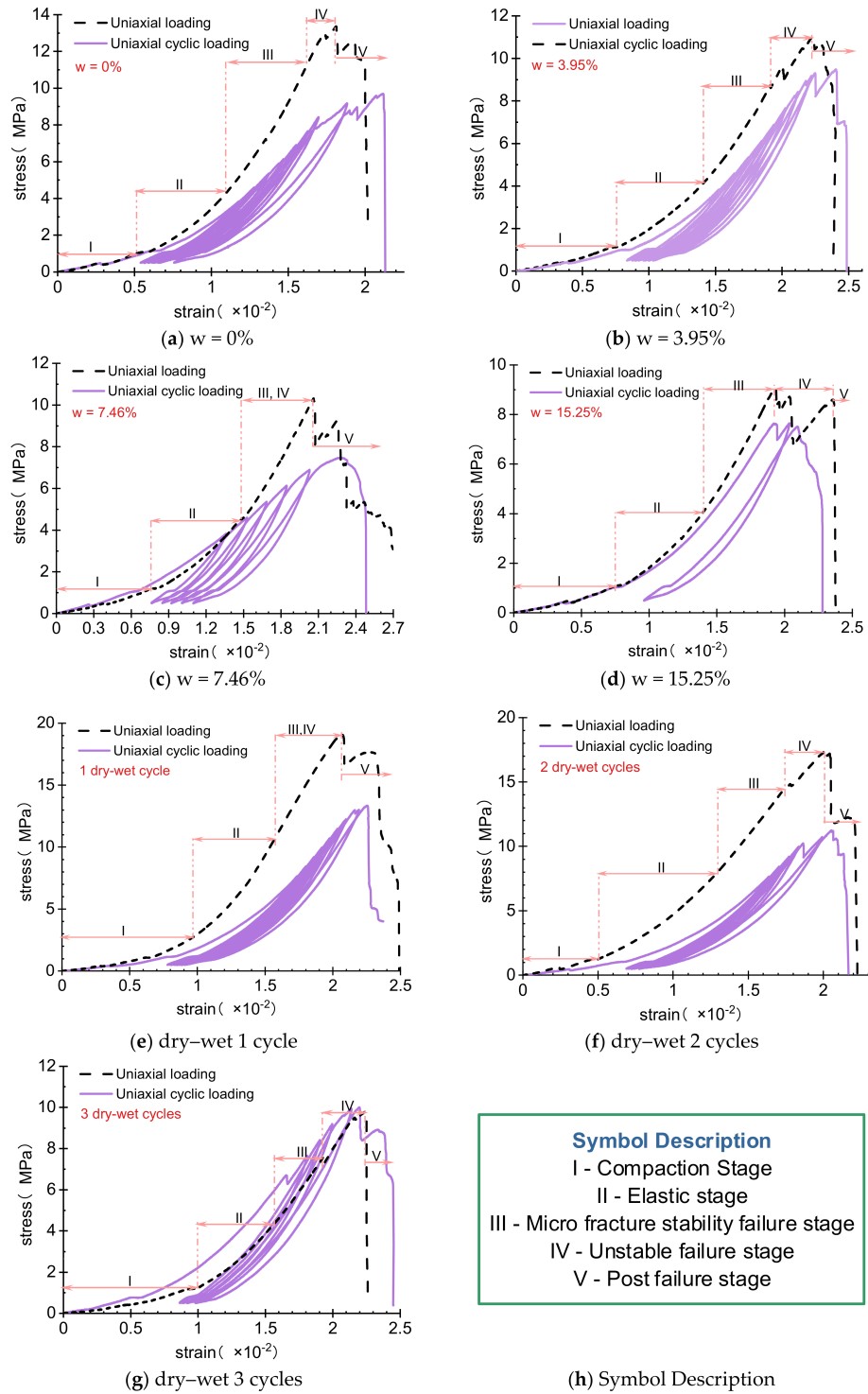

**Figure 6.** Different moisture state coal sample stress-strain curve.

As shown from Figure 6a–d, the stress-strain characteristics of the uniaxial compression conditions of coal samples with different moisture contents can be divided into five stages, as follows: compaction stage, elastic stage, micro-fracture stability failure stage, unstable failure stage, and post-failure stage. The peak stress and peak strain of dry coal samples (i.e., coal samples with the moisture content of 0%) under uniaxial compression load is 13.36 MPa and $1.71 \times 10^{-2}$, whereas that under cyclic compression load is 9.69 Mpa a is $2.11 \times 10^{-2}$. The peak stress under the cyclic compression load is 72.53% of that under the uniaxial compression load, but the peak strain under the cyclic compression load is higher than that under the uniaxial compression load. As presented in Figure 6a–d, the strain peak on the stress-strain curves in the cyclic loading–unloading test lags that in the uniaxial compression test. These curves are concave and have a high coincidence in the compaction stage and elastic stage.

It indicates that at the initial stage of stress loading, initial pores and fractures in dry coal samples are gradually compacted and closed, and the stress increases rapidly with the increase in loading displacement. In the stage of micro-fracture stable failure, with the increase in loading cycles, the stress-strain curve during the cyclic loading–unloading test gradually deviates downward from that in the uniaxial compression test, and the hysteresis loop area gradually increases. In the unstable failure stage, the coal sample in the uniaxial compression test loses stability and fails rapidly after reaching the peak stress, whereas the coal sample in the cyclic loading–unloading test has a long-term instability process with the increase in cyclic loading–unloading. After the $9^{th}$ cyclic loading–unloading, the coal sample is destroyed.

When the moisture content is 7.46%, most of the stress-strain curves of the cyclic loading–unloading test are still below those of the uniaxial compression test. After the sixth cyclic loading–unloading, the coal sample is destroyed. Different from the uniaxial compression test, the stress-strain curve of the coal sample in the cyclic loading–unloading test has no obvious post-peak stage.

As shown from Figure 6e–g, the peak stress and peak strain of the coal sample after 1 dry–wet cycle in the uniaxial compression test are 19.08 Mpa and $2.07 \times 10^{-2}$, whereas those in the cyclic loading–unloading test are 13.34 Mpa and $2.25 \times 10^{-2}$. It shows that in the cyclic loading–unloading test, the compressive strength of coal samples decreases, the peak stress is 70.04% of that in the uniaxial compression test, and the peak strain is slightly higher than that in the uniaxial compression test. Under cyclic compression load, the coal sample loses stability immediately after crack propagation and penetration; whereas under uniaxial compression load, it has an obvious post-peak stage.

The peak stress of coal samples after 3 dry–wet cycles under cyclic compression load is like that under uniaxial compression load, but the peak strain of coal samples under cyclic compression load is relatively small. In addition, there is no obvious post-peak stage of coal samples under uniaxial compression load, and the stress–strain curve under cyclic compression load has an obvious post-peak stage.

### 3.2. Mechanical Parameter of Water-Bearing Coal under Cyclic Loading

The compressive strength, peak strain, and elastic modulus of coal samples are important parameters for studying the mechanical properties of coal [34]. In order to reduce the experimental error, the average values of compressive strength and peak strain under cyclic loading–unloading are used, and the elastic modulus in the unloading curve in each cycle is used. Tables 2–4 show the detailed parameters of coal samples.

**Table 2.** Mechanical parameters of coal samples with different moisture contents under cyclic loading–unloading.

| W (%) | $\sigma_{max}$ (Mpa) | $\sigma_{max}{'}$ (Mpa) | $\varepsilon_{max}$ ($\times 10^{-2}$) | $\varepsilon_{max}{'}$ ($\times 10^{-2}$) | E (Gpa) | E′ (Gpa) |
|---|---|---|---|---|---|---|
| 0 | 13.36 | 9.69 | 1.71 | 2.11 | 1.22 | 1.05 |
| 3.95 | 10.99 | 9.46 | 2.22 | 2.39 | 0.74 | 0.97 |
| 7.46 | 10.30 | 7.44 | 2.29 | 2.31 | 0.63 | 0.95 |
| 15.25 | 9.11 | 7.64 | 1.94 | 2.04 | 0.56 | 0.88 |

**Table 3.** Mechanical properties of coal samples after different dry–wet cycles under cyclic loading–unloading.

| Dry–Wet Cycles | $\sigma_{max}$ (Mpa) | $\sigma_{max}{'}$ (Mpa) | $\varepsilon_{max}$ ($\times 10^{-2}$) | $\varepsilon_{max}{'}$ ($\times 10^{-2}$) | E (Gpa) | E′ (Gpa) |
|---|---|---|---|---|---|---|
| 1 | 19.08 | 13.34 | 2.07 | 2.25 | 1.51 | 1.37 |
| 2 | 17.34 | 11.23 | 2.00 | 2.05 | 0.98 | 1.40 |
| 3 | 9.79 | 10.00 | 2.50 | 2.20 | 0.56 | 1.24 |

**Table 4.** Statistical table of elastic modulus of coal samples in different water-bearing states under cyclic loading and unloading.

| Water-Bearing State | Elastic Modulus of Each Cycle Loading (Mpa) | | | | | | | |
|---|---|---|---|---|---|---|---|---|
| | 1 | 2 | 3 | 4 | 5 | 6 | 7 | 8 |
| $w = 0\%$ | 1192.75 | 1100.90 | 1051.66 | 1043.79 | 1032.91 | 1023.10 | 997.63 | 996.62 |
| $w = 3.95\%$ | 1067.15 | 1016.32 | 999.10 | 987.95 | 967.31 | 960.13 | 903.19 | 880.56 |
| $w = 7.64\%$ | 1030.57 | 1020.54 | 1017.25 | 844.17 | 825.00 | | | |
| $w = 15.25\%$ | 925.54 | 821.72 | | | | | | |
| 1 dry–wet cycle | 1186.26 | 1197.42 | 1273.36 | 1291.79 | 1285.47 | 1596.83 | 1620.82 | 1748.02 |
| 2 dry–wet cycles | 1270.55 | 1254.42 | 1370.17 | 1469.69 | 1499.80 | | | |
| 3 dry–wet cycles | 1109.43 | 1216.62 | 1300.85 | 1315.02 | | | | |

As shown in Figure 7, the determination of elastic modulus $E'$ in each cycle refers to the determination method proposed by Pourhosseini and Shabanimashcool [35], that is, the area enclosed by the elastic modulus $E'$ straight line and the strain axis is equal to the area enclosed by the unloading curve and the strain axis of each cycle. The elastic modulus of each cycle can be obtained by making a straight line representing the elastic modulus of the same area.

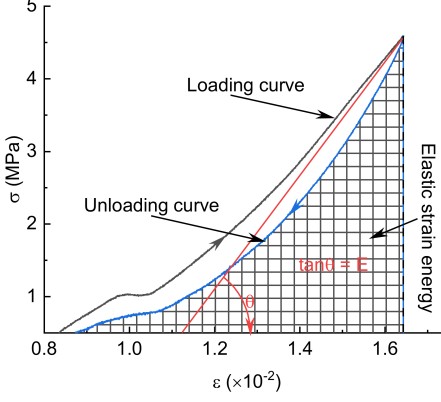

**Figure 7.** Calculation method of elastic modulus under cyclic loading–unloading.

As shown in Figure 8a,c,e, the peak stress of the dry coal sample is the highest, which is 9.69 Mpa. With the increase in moisture content, the peak stress and peak strain decreased

to varying degrees. From the microscopic point of view, water molecules enter the micro-cracks and pores in the coal body and adhere to the inner surface of the coal to form bound water. Under the action of external load, part of the bound water expands, promotes the expansion of the cracks in the coal, reduces the bearing capacity of the coal, and accelerates its damage.

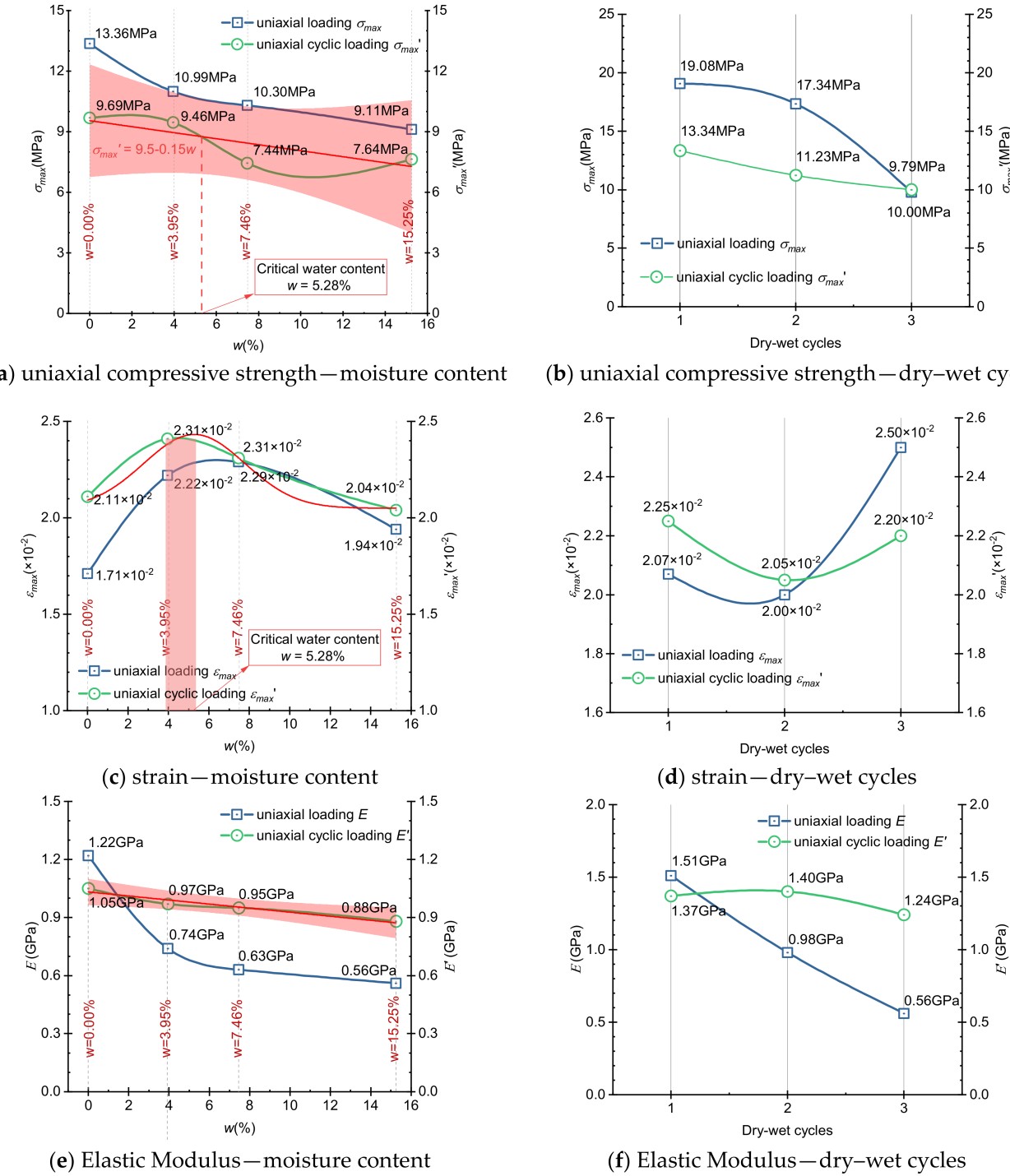

**Figure 8.** Evolution of uniaxial loading and cyclic loading mechanical parameters of coal samples with different water content states, (the red area represents the confidence interval of the fitted straight line).

It is worth noting that when the moisture content increases from 3.95% to 7.46%, the changing range of compressive strength and peak strain is the largest. Specifically, the compressive strength decreases from 9.46 MPa to 7.44 Mpa, which decreases by 21.20%; the peak strain decreases from $2.39 \times 10^{-2}$ to $2.31 \times 10^{-2}$, which decreases by 3.33%. This is because when the moisture content reaches a certain value, most of the water molecules have been filled into the natural fractures; under the external pressure of cyclic loading, the fracture expansion effect is the strongest, and the macroscopic performance is that the compressive strength of coal body decreases sharply. Therefore, there is an inflection point in the strength attenuation of coal samples under cyclic loading, that is, the critical moisture content. It is speculated that when w = 5.28% as indicated in the figure, the strength attenuation of coal samples is the largest.

As shown in Figure 8b,d,f, with the increasing number of dry–wet cycles, the peak stress and peak strain of the coal sample after dry–wet cycles treatment decrease under the action of cyclic compression load. After three dry–wet cycles, the peak stress of the coal sample decreases from 13.34 Mpa to 10.00 Mpa, decreased by 24.72%, and the peak strain decreases from $2.25 \times 10^{-2}$ to $2.20 \times 10^{-2}$, reduced by 0.5%. Therefore, the dry–wet cycle treatment of coal samples has a great impact on the weakening of coal strength.

Figure 9a depicts the changing trend of elastic modulus of coal samples with different moisture contents under different numbers of cyclic loading–unloading. The change of elastic modulus in the first three cyclic loading–unloading of dry coal samples is the largest. The elastic modulus decreases from 1192.75 Mpa to 1051.66 Mpa in the first cyclic loading–unloading, and then slowly decreases to 996.62 Mpa in the second cyclic loading–unloading and then tends to be stable. The change of elastic modulus of coal samples with the moisture content of 3.95% is the largest in the first and sixth cyclic loading–unloading, decreasing 50.83 Mpa and 56.94 Mpa respectively. The coal sample with the moisture content of 7.46% will be unstable and damaged after 5 loading–unloading cycles. The change of elastic modulus is the largest in the third cyclic loading–unloading, which is reduced by 173.08 Mpa. Therefore, the higher the moisture content, the smaller the elastic modulus at the sample failure, and the elastic modulus decrease gradually with the increasing number of cyclic loading–unloading.

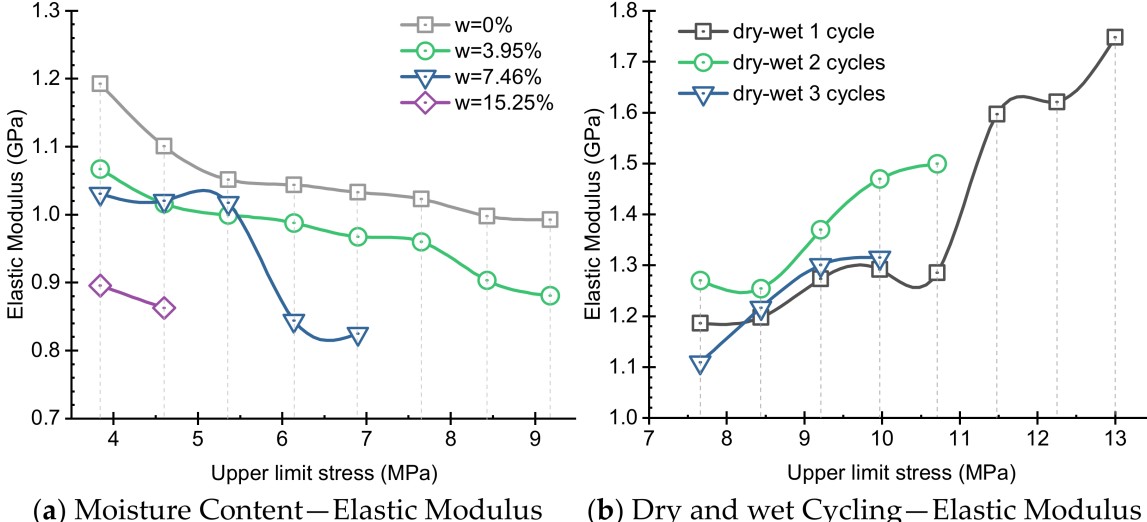

(**a**) Moisture Content—Elastic Modulus  (**b**) Dry and wet Cycling—Elastic Modulus

**Figure 9.** Relationship between upper limit stress and elastic modulus of coal samples with different water content.

Figure 9b shows the variation law of elastic modulus of coal samples after different dry–wet cycles. The elastic modulus of coal samples after dry–wet cycles is positively correlated with the number of dry–wet cycles, that is, the elastic modulus increases with the increasing number of dry–wet cycles. This shows that under the action of cyclic

compression load, the plasticity of coal samples is gradually enhanced by repeated soaking and drying treatment. In addition, the coal samples treated by one dry–wet cycle are unstable and damaged after the eighth cyclic loading–unloading, and its elastic modulus is 1748.02 MPa at this time. The coal samples treated by 2 and 3 dry–wet cycles failed at the fifth and fourth cyclic loading–unloading, and the elastic modulus of these samples are 1499.80 Mpa and 1315.02 Mpa, respectively. The more dry–wet cycles on the coal samples, the lower the elastic modulus of coal samples.

### 3.3. Failure Characteristics of Water-Bearing Coal Samples under Cyclic Loading–Unloading

As shown in Figure 10a–d, under the action of cyclic load, with the increase in moisture content, the position of fracture development shifts from the edge to the interior., the number of secondary fractures around the main fracture decreases gradually, and the failure mode changes from shear failure to shear-tensile failure. This is because water gradually invades the coal sample from outside to inside. When the moisture content is low, water hardly penetrates the coal sample and is mainly located at the edge, that is, the strength at the edge is clearly weakened and longitudinal cracks are mostly generated. With the increase in moisture content, the cracks begin to expand, and gradually stabilize when the crack failure limit is exceeded. When the crack density reaches the level when the cracks condense into shear bands or stretch and peel off the coal sample [36].

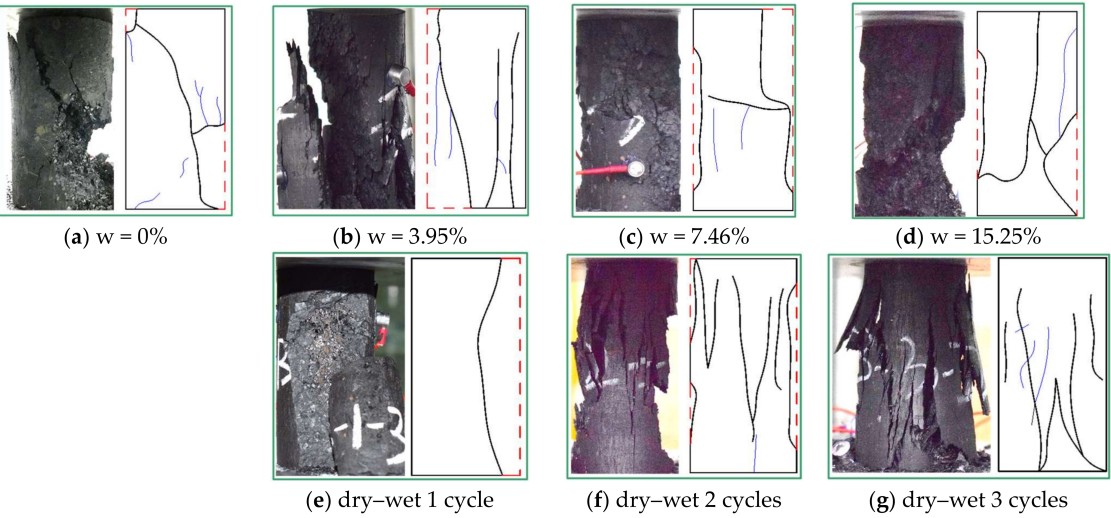

(**a**) w = 0%   (**b**) w = 3.95%   (**c**) w = 7.46%   (**d**) w = 15.25%

(**e**) dry–wet 1 cycle   (**f**) dry–wet 2 cycles   (**g**) dry–wet 3 cycles

**Figure 10.** Failure characteristics of coal samples after different dry–wet cycles under cyclic loading–unloading.

As shown in Figure 10e–g, a main longitudinal crack is generated in the coal sample after one dry–wet cycle under cyclic loading–unloading conditions. This longitudinal crack runs through the entire coal sample. The stripped coal body is massive and has high integrity. With the increasing number of dry–wet cycles, the length of the fracture decreases, but the number of fractures increases. These fractures are approximately V-shaped, and evenly and symmetrically distributed about the axis of the coal sample. The stripped coal body gradually decreases, changing from block to debris; the integrity of the coal sample is enhanced. The fracture morphology of coal samples is consistent with the various characteristics of mechanical parameters under cyclic loading–unloading. The more dry–wet cycles on coal samples, the more significant the plastic failure.

### 3.4. Acoustic Emission Characteristics of Water-Bearing Coal under Cyclic Loading

As shown in Figure 11, where (a), (b), (c) and (d) represent water content of 0, 3.95%, 7.46% and 15.25% of coal samples. The transformation from blue area to red area represents the distribution density from small to large. From the probability density diagram, the core areas of the three kinds of water-bearing coal samples are all located at the upper

left of the tension-shear boundary, indicating that the penetration of tension cracks under cyclic loading finally leads to the macroscopic failure of coal samples. According to the observation of the failure characteristic diagram of coal samples, with the increase in water content, the shear crack decreases gradually, and the dominant role of tension crack increases gradually. So, the RA-AF density map has limitations in evaluating fracture types in coal samples of different moisture contents subjected to cyclic loading conditions.

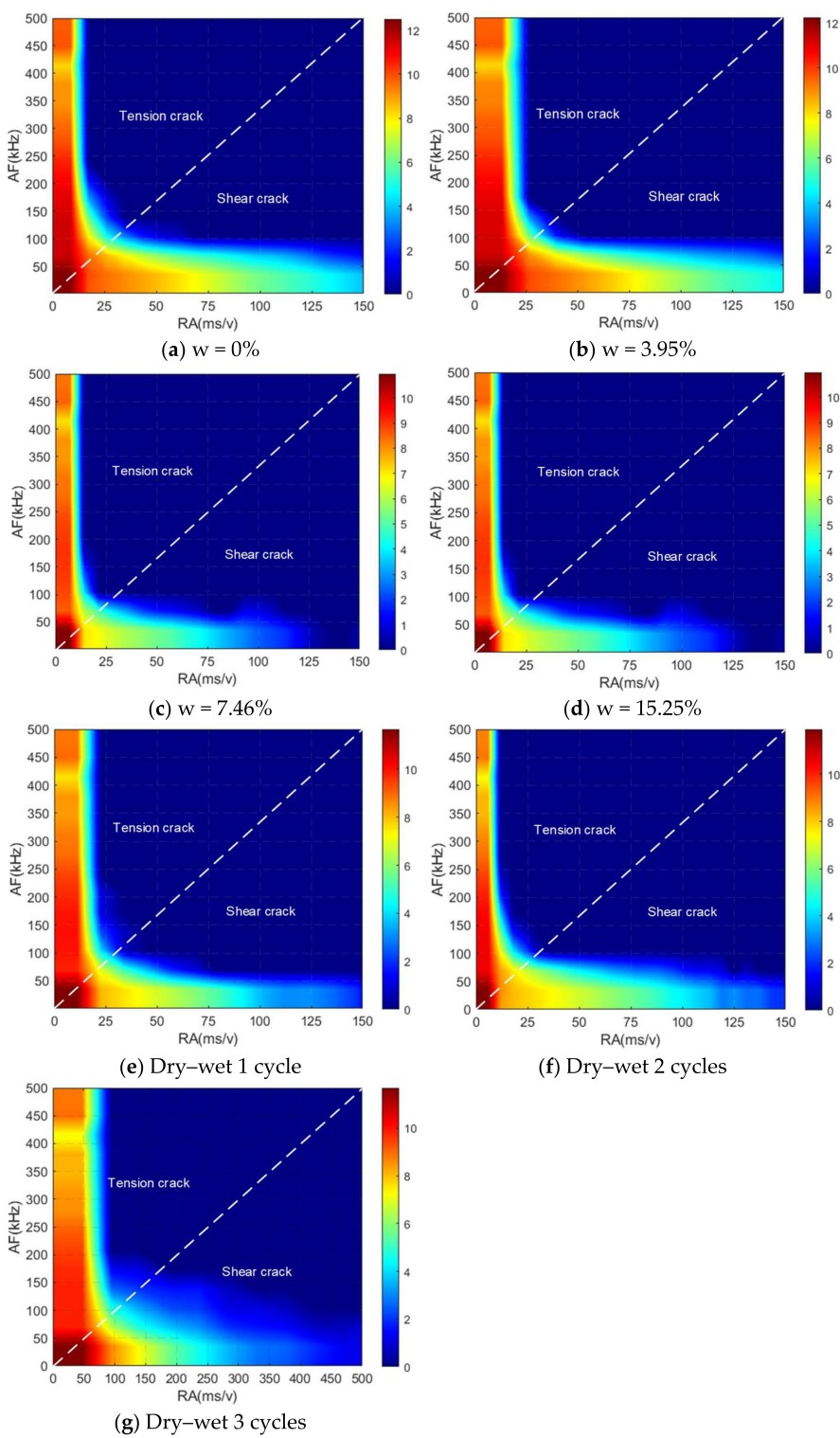

**Figure 11.** RA-AF probability density distribution map.

From Figure 11e–g, the core area of the damaged fissures of coal samples treated with different numbers of wet and dry cycles are all located at the upper left of the tension-shear dividing line, indicating that the penetration of tension fissures under the cyclic loading eventually leads to the macroscopic damage of coal samples. With the increase in the number of dry and wet cycle treatments, both tension fractures and shear fractures gradually increase, and the dominant role of tension fractures gradually increases.

The fracture evolution and failure characteristics of water-bearing coal samples can be explained in combination with the water weakening mechanism and pore mechanics. Firstly, water enters the internal lattice, weakening the bonds of the original structure inside coal samples, which changes the coal microstructure and reduces its cohesion. It leads to the decrease of pore connectivity, and the increase in pore pressure in pores with poor connectivity will cause the rotation of stress direction and lead to tensile failure under cyclic loading [37].

## 4. Conclusions

In this study, the uniaxial cyclic loading–unloading experiments of coal samples with different moisture contents and dry–wet cycles were performed under acoustic emission and photographic monitoring, and the variation law and failure characteristics of mechanical properties of coal samples were studied. The main conclusions are as follows:

(1) The loading mode has a significant influence on the strength and deformation characteristics of water-bearing coal samples. The peak stress of coal samples under cyclic loading–unloading is about 80% of that under the uniaxial loading, and the peak stress gradually decreases with the increase in moisture content and dry–wet cycles. With the increase in moisture content and dry–wet cycles, the hysteresis loop of coal samples changes from a dense state to a sparse state. With the increasing number of dry–wet cycles, the resistance to fatigue damage increases and is inversely proportional to the moisture content, and the stress-strain curve of coal samples appears post-peak stage, and the plasticity of coal samples is enhanced.

(2) Water has a significant effect on mechanical parameters such as elastic modulus. Under cyclic load, when the moisture content is 5.28%, the mechanical parameters (such as strength and elastic modulus) of the coal sample decrease the most. Under the condition of moisture content, the elastic modulus of the coal sample is inversely proportional to the number of the cyclic loading–unloading. The higher the moisture content, the greater the decreasing rate of elastic modulus. After the dry–wet cycle treatment, the elastic modulus of the coal sample is directly proportional to the number of wet–dry cycles, and the more dry–wet cycles, the smaller the increasing rate of elastic modulus.

(3) The fracture development characteristics are controlled by the moisture content of coal samples. With the increase in moisture content, the position of fracture development shifts from the edge to the interior, the number of secondary fractures around the main fracture decreases gradually, and the failure mode changes from shear failure to shear-tensile failure. After one dry–wet cycle treatment, the coal sample produces a main longitudinal crack, which runs through the sample. The stripped coal body is massive and has high integrity. With the increase in dry–wet cycles, the length of fracture decreases, but the number of fractures increases. These fractures are approximately V-shaped, evenly, and symmetrically distributed about the axis of the coal sample. The stripped coal body gradually decreases, changing from block to debris, and the integrity of the coal sample is enhanced.

(4) From the probability density diagram, the penetration of tension cracks under cyclic loading finally leads to the macroscopic failure of coal samples. For evaluating fracture types, the RA-AF density map is more applicable to wetting–drying cycle coal samples, whereas for the coal samples with different moisture contents should be carried out with caution. With the increase in water content, the shear crack decreases gradually, and the dominant role of tension crack increases gradually. With the increase in the number of dry and wet cycle treatments, both tension fractures and shear fractures gradually increase, and the dominant role of tension fractures gradually increases.

**Author Contributions:** C.S.: investigation. H.X.: methodology. L.Y.: data curation. H.X.: writing and original draft preparation. Q.Y.: writing, review and editing. All authors have read and agreed to the published version of the manuscript.

**Funding:** The authors gratefully acknowledge financial support from the National Natural Science Foundation of China (No. 51874283).

**Institutional Review Board Statement:** Not applicable.

**Informed Consent Statement:** Not applicable.

**Data Availability Statement:** Not applicable.

**Acknowledgments:** The authors gratefully thank the anonymous reviewers for their constructive comments on improving the presentation. All authors have agreed to the listing of authors.

**Conflicts of Interest:** The authors declare they have no financial interests.

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
