# Peer review of "Study on Damage Characteristics of Water-Bearing Coal Samples under Cyclic Loading–Unloading"

_sustainability, doi:10.3390/su14148457_

Round 1
Reviewer 1 Report
This work represents well for the readers, and it can be accepted for publication if the authors can improve the manuscript with some minor revision. Of course, there are many similar studies that are reported by the previous researchers.
The Chinese map is incomplete, please pay special attention to Fig. 2 in the manuscript, or that the study should be rejected by the reviewers, or everyone in China. by the way, change the red word into black, and keep the same type size.
line 132, 133, 348, Error! Reference source not found., line 135, eq. (1) is missing
The effect of the bedding plane in the coal samples should be taken into consideration in the disscussion section. and please metion it in the section 2, how to prepare the coal samples.
eg. International Journal of Rock Mechanics and Mining Sciences, 2019, 117: 49-62.
And the data analysis method can be used for reference
eg. Engineering Fracture Mechanics, 2021, 245: 107581.
Reviewer 2 Report
The manuscript is a study about damage characteristics of water-bearing coal samples under cyclic loading-unloading. The scope of this article is consistent with the requirements of the Sustainability, but it requires minor revision in accordance with the comments below:
1. The abstract is too long. According to the Instructions for Authors the abstract should be a total of about 200 words maximum.
2. Avoid linking references as in lines: 41-42, 62, 77. Please combine no more than two references. Instead summarise the main contribution of each referenced paper in a separate sentence.
3. References must be numbered in order of appearance in the text and listed individually at the end of the manuscript.
4. Section 2 should be Materials and Methods.
5. Section 3 should be Results and Discussion.
6. The lines 327-347 should be moved to section Materials and Methods.
Round 2
Reviewer 1 Report
The authors revised the manuscript carefully, except for line 320, 428, Error! Reference source not found, the study can be accepted for publication, and before this, the abstract can be polished by adding much more highlights and key points.
